# Plasma Proteomic Analysis Reveals the Potential Role of Lectin and Alternative Complement Pathways in IgA Vasculitis Pathogenesis

**DOI:** 10.3390/diagnostics13101729

**Published:** 2023-05-12

**Authors:** Selcan Demir, Idil Yet, Melis Sardan Ekiz, Erdal Sag, Yelda Bilginer, Omur Celikbicak, Incilay Lay, Seza Ozen

**Affiliations:** 1Department of Pediatric Rheumatology, Hacettepe University Medical Faculty, 06230 Ankara, Turkey; 2Department of Bioinformatics, Graduate School of Health Sciences, Hacettepe University, 06230 Ankara, Turkey; 3Advanced Technologies Application and Research Center (HUNITEK), Hacettepe University, 06230 Ankara, Turkey; 4Department of Chemistry, Hacettepe University, 06230 Ankara, Turkey; 5Department of Biochemistry, Hacettepe University Medical Faculty, 06230 Ankara, Turkey

**Keywords:** Henoch–Schönlein purpura, IgA vasculitis, pathogenesis, proteomic, lectin complement pathway, alternative complement pathway

## Abstract

Background: IgA vasculitis (IgAV) is the most common form of childhood vasculitis. A better understanding of its pathophysiology is required to identify new potential biomarkers and treatment targets. Objective: to assess the underlying molecular mechanisms in the pathogenesis of IgAV using an untargeted proteomics approach. Methods: Thirty-seven IgAV patients and five healthy controls were enrolled. Plasma samples were collected on the day of diagnosis before any treatment was initiated. We used nano-liquid chromatography–tandem mass spectrometry (nLC–MS/MS) to investigate the alterations in plasma proteomic profiles. For the bioinformatics analyses, databases including Uniprot, PANTHER, KEGG, Reactome, Cytoscape, and IntAct were used. Results: Among the 418 proteins identified in the nLC–MS/MS analysis, 20 had significantly different expressions in IgAV patients. Among them, 15 were upregulated and 5 were downregulated. According to the KEGG pathway and function classification analysis, complement and coagulation cascades were the most enriched pathways. GO analyses showed that the differentially expressed proteins were mainly involved in defense/immunity proteins and the metabolite interconversion enzyme family. We also investigated molecular interactions in the identified 20 proteins of IgAV patients. We extracted 493 interactions from the IntAct database for the 20 proteins and used Cytoscape for the network analyses. Conclusion: Our results clearly suggest the role of the lectin and alternate complement pathways in IgAV. The proteins defined in the pathways of cell adhesion may serve as biomarkers. Further functional studies may lead the way to better understanding of the disease and new therapeutic options for IgAV treatment.

## 1. Introduction

IgA vasculitis (IgAV), formerly named Henoch–Schönlein purpura (HSP), is the most common form of childhood vasculitis. In 2012, Chapel Hill Consensus replaced the eponym HSP with IgAV since the hallmark of the pathogenesis is the abnormal IgA-dominant immune deposits in small vessel walls [1]. The classical presentation of IgAV is palpable purpura, arthralgia/arthritis, and gastrointestinal involvement. Glomerulonephritis, which is histopathologically indistinguishable from IgA nephropathy, may also occur in 1/3 of patients. The symptoms in the acute phase of the disease resolve spontaneously in the vast majority of children. However, renal involvement can persist and cause end-stage renal disease [2].

Despite the fact that IgAV is the most common childhood vasculitis that pediatricians encounter in daily practice, there are many unanswered questions about the pathophysiology of the disease. The main component of vascular inflammation is IgA1-dominant immune deposits and it has been also shown that IgA1 in IgAV is galactose-deficient (Gd-IgA1) [3]. A multi-hit hypothesis has been proposed to explain the role of Gd-IgA1 in IgAV and IgAVN [4]. This hypothesis suggests that Gd-IgA1 may share mimicry with certain microbes and increased levels of Gd-IgA1 triggered by environmental and/or that genetic factors are recognized by autoantibodies. Immune complexes containing Gd-IgA1 accumulated in the mesangium are responsible for kidney damage; however, their role in systemic inflammation in IgAV remains controversial. An alternative multi-hit hypothesis suggests that cross-reactive IgA1 antiendothelial cell antibodies, also produced as a result of molecular mimicry, initiate systemic inflammation by binding to endothelial cells, leading to neutrophil infiltration and vascular damage [5]. It is still unknown why this abnormal IgA1 glycosylation occurs and what the concomitant and/or subsequent inflammatory processes and pathways involved are.

Since the pathogenesis has not fully been elucidated, we still lack evidence-based treatment approaches for the management of IgAV. “Omics” has been widely used in exploring the underlying mechanisms of disease development. Recent reports have investigated omics in IgA vasculitis [6,7,8,9]. We published a metabolomic study that revealed potential biomarkers associated with the progression of IgAV to IgAV with nephritis [10]. Combining the results of different “omics” techniques with big data analysis, would provide better understanding of the possible unknown mechanisms underlying the pathophysiology of the disease and guide us in personalized treatment. In this study, we performed a nano-liquid chromatography–tandem mass spectrometry-based untargeted proteomics approach to elucidate the pathways involved in the pathogenesis of IgAV.

## 2. Materials and Methods

### 2.1. Patient Selection and Sample Collection

Patients are diagnosed with IgAV according to the Ankara 2008 criteria endorsed by the European League Against Rheumatism, Pediatric Rheumatology International Trials Organization, and Pediatric Rheumatology European Society (EULAR/PRINTO/PRES) [11]. Thirty-seven active IgAV patients, and five age-matched healthy controls were enrolled in this study. Informed written consent was obtained from parents of subjects. Blood samples were collected on the same day of IgAV diagnosis before any medication was initiated. All patients had a minimum of 1-year follow-up after diagnosis and had no other systemic vasculitis or immunological diseases. Ethical approval was obtained from the Institutional Review Board (GO-18/81-08). Details on sample preparation and nano-liquid chromatography–tandem mass spectrometry (nLC–MS/MS) analysis are provided in the Appendix A.

### 2.2. Protein Identification and Bioinformatic Analyses

MaxQuant (version 1.16.14) coupled with the Andromeda search engine was used to identify the proteins in the Uniprot database (UP000005640) and supplemented with frequently observed contaminants [12]. Broker Q-TOF instrument was selected with its default parameters and a Trypsin/P specificity allowing for 2 missed sites were set as search parameters. Fixed modifications of carboxyamidomethyl (C) and variable modifications of oxidation (M), acetyl (protein N-term) were set. A minimal peptide length of seven amino acids was set. MaxQuant was used to perform an internal mass calibration of measured ions and peptide validation by way of the target decoy approach. Proteins and peptides with 1% false discovery rate were accepted if they had been identified by at least 1 peptide in one of the samples. Peptide sequences were selected as razor and unique peptides for the quantification. Proteins were computed as label free quantification (LFQ) values in MaxQuant following normalization [12].

### 2.3. Protein Quantification Analysis and Functional Annotation Analysis

LFQ values were log2-transformed using Perseus software suite 1.6.7 to achieve normal distribution [13]. Proteins identified in at least three runs and LFQ values equal to zero were kept. Statistical significance of changes in abundance between case and control groups was calculated using a two-tailed *t*-test, with *p*-values adjusted using the Benjamini–Hochberg method. R programming language (R version 4.0.4) was used to calculate fold changes (FC). A protein is considered significant if meets the log2 FC ≥ 1.5 and log2 FC ≤ −1.5 with adjusted *p*-value < 0.05 criteria. Principal component analysis (PCA) was used, which allows for natural clustering behavior to be visualized in the context of sample groups using all significant proteins. For this, first, two PCs (principal components) were plotted against each other. Differentially expressed proteins were visualized in a heatmap using hierarchical clustering analysis using default Euclidean distance. Proteins were functionally categorized using Gene Ontology (GO) system by way of PANTHER classification system based on biological processes, molecular activity, and cellular components against a background of the Homo sapiens proteome [14]. Pathway analysis and functional interactions were performed using the Kyoto Encyclopedia of Genes and Genomes (KEGG) and Reactome database [15,16]. The functional networks for the significant proteins and their interactions were generated through the use of the Cytoscape database [17]. The interactions for the significant proteins were downloaded from the IntAct database [18].

## 3. Results

### 3.1. Demographic and Clinical Characteristics

A total of 37 patients diagnosed with IgAV (24 females, 13 males, mean age 8.77 ± 3.54 years) and 5 healthy children serving as the control group (3 males and 2 females, mean age 8.91 ± 4.07), were enrolled in this study. At the time of diagnosis, all (100%) patients had skin involvement, 25 (67%) had arthritis, and 15 (40%) had gastrointestinal involvement. None of those patients developed renal involvement at the time of diagnosis or one-year follow-up. Among males, one (7.6%) also had scrotal involvement.

### 3.2. Proteomic Profiling

Four hundred and eighteen proteins identified in three fractions were quantified using label-free quantification (LFQ) and transformed into relative expression data resulting in expression profiles for all 418 proteins in the plasma from patients with IgAV and healthy children. Three hundred forty-eight proteins were used for downstream analysis, removing contamination, reverse effect, and only identified by one-site proteins. Twenty-five of them were found to be statistically different compared with those from healthy controls. Among 20 proteins, 15 were upregulated and 5 were downregulated.

In order to obtain a simplified structure of the behavior of all identified proteins under two groups, we subjected the significant protein LFQ intensities to PCA and heatmap clustering. PCA projection demonstrates that healthy children and IgAV patients clustering together that shows the maximum variability in the dataset occurs between groups with the first component covering 60.6% of the data variance (Figure 1A). This result is also reflected in the heatmap, where two major clusters separating the significant protein LFQ intensities in two groups as IgAV patients and healthy controls (Figure 1B).

### 3.3. KEGG and Reactome Pathway Analysis of the Differentially Expressed Proteins

KEGG pathway analysis includes 13 of 20 differentially expressed proteins and complement and coagulation cascades were the most enriched pathways. (Table 1, Figure 2A). Additionally, the Reactome pathway analysis identified 17 out of 20 differentially expressed proteins (Appendix A). Pathways of genes involved in cell surface interactions at the vascular wall (FXIIIA1 and complement 9), the one involved in fibrin clots (complement factor B)—involved in VEGF signaling—were clearly demonstrated (Appendix A).

Genes involved in the complement cascade including glycosylphosphatidylinositol-specific phospholipase D1 (GPLD1), carnosine dipeptidase 1 (CNDP1) (in KEGG only), superoxide dismutase 3 (SOD3), and butyrylcholinesterase (BCHE) were also prominent in this immune complex disease (Figure 2A and Appendix A).

### 3.4. Panther Classification Analysis and GO Functional Analysis of the Differentially Expressed Proteins

To further understand the molecular and biological functions of these identified proteins, 20 significant proteins were also classified based on GO using the PANTHER classification system. GO analysis revealed the biological processes, cellular components, and molecular functions of the differentially expressed proteins that are implicated in IgAV patients. The genes, overexpressed among the IgAV patients, were mostly involved in cellular processes (9 proteins) under biological processes, binding (7 proteins) under molecular function, and cellular anatomical entity (14 proteins) under cellular component (Figure 2B and Appendix A). PANTHER protein classification analysis showed that the differentially expressed proteins were mainly involved in defense/immunity proteins and metabolite interconversion enzyme family.

We investigated molecular interactions in the identified 20 proteins and extracted 493 interactions from the IntAct molecular interaction database for all proteins. We performed PPI (protein–protein interaction) network analysis with these interactions and differentially expressed proteins using Cytoscape (Figure 3). The PPI network revealed the interaction between 17 out of 20 proteins mediated by other proteins. This finding may indicate that there are other genes that play an important role in IgAV.

## 4. Discussion

In this study, a Q-TOF LC/MS-based untargeted plasma proteomics approach was applied to identify novel pathways involved in the pathogenesis of IgAV. Although IgAV is the most common form of childhood vasculitis, there are still significant knowledge gaps in its pathogenesis. We defined 20 proteins that were significantly expressed in IgAV, with 15 proteins being upregulated and 5 being downregulated. It was remarkable that one of the prominent pathways was the complement pathway, while we also observed upregulated proteins involved in cellular adhesion and coagulation cascades with altered fibrin structure. We address these pathways separately:

I. The complement pathway: KEGG function classification analysis demonstrated that the complement cascades were indeed the most enriched pathways. In the complement pathway, mannose-binding lectin-associated serine protease-1 (MASP1) exhibited a 13.46-fold increase in IgAV. MASP1 functions in the lectin pathway of complement activation and amplifies complement activation by cleaving complement C2 or by activating another complement serine protease, MASP2 (Figure 4). Cleavage of C4 and C2 with MASP1 and MASP2 forms C3 convertase and C5 convertase. Ultimately, the generated C3a and C5a from the lectin, as well as the classical and alternative pathways of complement activation, induce a potent inflammatory response and stimulate the recruitment of peripheral immune cells. Finally, activation of the complement system leads to a robust local and systemic inflammatory response [19]. In our study, the increased MASP1 in IgAV suggests a significant stimulation in the lectin pathway of complement activation. Moreover, the 3.7-fold decrease observed in the serine protease inhibitor (SERPINA5) in IgAV may further boost the function of MASP1 together with MASP2 in complement activation, since MASPs are serine proteases. IgAV manifests as a systemic small vessel vasculitis that may involve the kidneys as IgA nephropathy (IgAN). There is growing evidence for the role of the complement pathways in IgAN and IgA vasculitis-associated nephritis (IgAVN) as well. Some patients with IgAN have been shown to have mesangial depositions of MASP1, MASP2, MBL, L-ficolin, and C4d indicating that the lectin pathway is activated. Roos et al. suggested that the deposition of MBL and other lectin pathway components is associated with more severe histopathological lesions and a worse outcome [20].

We also observed a 1.72-fold increase in complement factor B (CFB) which is a part of the alternative pathway of the complement system. Increased CFB suggests the contribution of the alternative pathway to complement activation and eventually to inflammatory and phagocytic response and cell lysis. The involvement of the alternative pathway was also suggested in a recent report [21]. Immunofluorescence studies revealed the colocalization of C3 with IgA1. Larger C3-immune complex deposits without an outer coat of IgA were also found to result in severe histological lesions [22,23,24].

The three known pathways of the complement system (lectin, alternative and classical) generate an active C3 convertase that creates C3a and C3b and finally shares the same sequence of events that results in the assembly of the membrane attack complex (MAC) [25]. It may be suggested that, in IgAV, the generation of C3a and C3b originates mostly from lectin, along with alternative pathways of complement activation. While C3a and C5a have inflammatory and chemoattractant activities through the corresponding C3a and C5a receptors, C5b initiates formation of the MAC. Subsequently, 10–16 units of C9 component are added to complex C5b-8, which is incorporated into the cell membrane creates a pore in the membrane, leading to cell lysis and death. We also found a 1.99-fold increase of C9 in IgAV. Overall, the high fold increases in MASP1 and CFB together with decreased SERPINA5 and increased C9 suggest the role of the lectin and alternate complement pathway in this form of vasculitis. The reported immunohistochemical findings of C3, properdin, C4d, MBL, and C5b-9 deposits in the mesangium of IgAN biopsy samples with the general absence of C1q support our results that the activation of alternative and lectin pathways are more prominent than the classical pathway [26]. Differential IgA glycosylation of monomeric and polymeric IgA bound in immune-complexes may have an impact on complement activation [27]. Glycosylation (complex N-glycans with terminal N-acetylglucosamine-galactose-deficient N-glycans) of IgG on its Fc part can also activate the lectin pathway [28].

To support the role of the complement pathway in IgAV, proteomic analysis of micro-dissected glomeruli from IgAN kidney biopsy samples have demonstrated significant amounts of C3, C5, and C6 to C9 located downstream through the complement activation system together with the accumulation of regulatory proteins of the alternative pathway [29]. The increased knowledge of the role of complement in the pathogenesis of IgAV has led to the consideration of new therapeutics targeting the complement system. In a recent case report, a young 21-year-old female with biopsy-proven IgAVN had failed to respond to corticosteroids and denied use of cyclophosphamide but stabilized when treated with narsoplimab, a monoclonal antibody used against MASP2 [30]. Subsequently, Patel et al. published a case report reporting the successful use of eculizumab, a humanized monoclonal antibody that inhibits cleavage of C5 by C5 convertase, in a 34-year-old female diagnosed with IgA vasculitis complicated by acute kidney failure [31]. There are already therapeutic agents targeting the complement activation system; one has been used in IgA nephropathy whereas another one has already been licensed for use in another small vessel vasculitis, the ANCA-associated vasculitides [30,32].

Increased levels (14.72-fold) of GPLD1 were also observed in IgAV. The biological processes among the broad processes of GPLD1 that may be associated with IgAV are the complement receptor-mediated signaling pathway and the regulation of endothelial cell migration [33]. High GPLD1 levels result in dysfunction of GPI-anchored complement regulators via degradation of GPI, and thus a more active complement system.

The most common feature of IgAV is palpable purpura, which reflects the small vessel vasculitis. Changes in coagulation function in IgAV are not obvious, though it has been considered by some authors [34,35]. We observed a fivefold decrease in coagulation factor XIII A chain (F13A1) in IgAV. Factor XIII is cleaved to F13A by thrombin, thus stabilizing the fibrin clot. In our recent metabolomics study, we showed increased levels of saccharopine in IgAV, an important amino acid in crosslinks between activated factor XIII and fibrin in clot formation [10]. Although patients with IgAV are known to have normal blood coagulation, our results indicated a hyperactivation of the coagulation system with an impaired fibrinolytic system and altered fibrin structure. More studies are needed to assess the dysregulation and the crossroad between the complement and the coagulation systems.

II. Proteins associated with the vasculature: KEGG analysis revealed that some of the differentially expressed proteins were associated with cell adhesion molecules such as cadherin-5, DNAH3, and IGFALS. Cadherin-5, also known as vascular endothelial cadherin, plays a role in endothelial adherence and junction assembly. Previous studies have reported increased circulating cadherin-5 levels in patients with Behçet disease and IgAV [36,37]. Chen et al. displayed increased serum levels of cadherin-5 in IgAV and showed that cadherin-5 correlated with the severity of the disease [38]. We found a 4.28-fold decrease in IGFALS, which is another cell adhesion molecule. Binding of IGFALS with IGF-1 increases vascular localization of IGF-1. In a mouse model, deficiency of IGFALS disrupts the circulation of IGF-1, which promotes vascular health [39]. IGF-1 also has anti-inflammatory effects and decreases the concentration of proinflammatory cytokines [40]. Thus, these proteins could be associated with vascular inflammation and vascular endothelium disturbances.

III. Others: REG3A, which had a 14.29-fold increase, involved in cell proliferation, differentiation, and innate immune response [41]. Darnaud et al. showed that, in a mouse model, intrarectal administration of REG3A decreases colonic inflammation [42]. It is tempting to speculate that REG3A may serve to suppress the gastrointestinal inflammation in IgAV.

A 12.95-fold increase was observed in LCAT, which catalyzes the hydrolysis of the platelet-activating factor (PAF) to lyso-PAF. PAF is the most potent plasmalogen, a well-recognized proinflammatory mediator resulting from membrane ether phospholipids. In our recent metabolomics study, increased alkyl-DHAP, which is the first committed intermediate in the synthesis of ether phospholipids–plasmalogens, was observed in IgAV [10]. These findings may suggest that phospholipids might have a role in the pathogenesis of IgAV.

Lysozyme-C increased 13.72-fold in IgAV. Lysozymes were found in myelocyte/macrophage cells within capillary loops and arterial walls in acute necrotizing vasculitis and have been considered as inflammatory markers [43,44].

An antioxidant enzyme SOD3 had a 13.69-fold increase and HYOU1 had a 10.28-fold increase in IgAV, which shows an oxidative stress probably associated with mitochondrial dysfunction and also cytoprotective cellular mechanisms triggered by oxygen deprivation.

We were not able to suggest substantial associations with some of the increased protein products. Further studies will enlighten their possible role in IgAV.

The major limitation of this study is the small number of the groups. Our study was an untargeted proteomic study, which is the first step in predicting biomarkers. Therefore, future studies are needed to validate our results in the plasma samples, possibly using LC–MS/MS-based targeted analytical techniques.

In conclusion, proteomic results may guide us not only in better understanding the disease but also in personalized treatment. We have shown, for the first time, using proteomic analysis, that the lectin and alternative complement pathways are involved in the pathogenesis of IgAV and that there is a prominent crossroad between complement and coagulation systems. Further functional studies may shed light on the role of the differentially expressed proteins defined in our study. Elucidating the role of activation of the complement or the other pathways defined in this study may identify new biomarkers and potential treatment targets in IgAV.

## Figures and Tables

**Figure 1 diagnostics-13-01729-f001:**
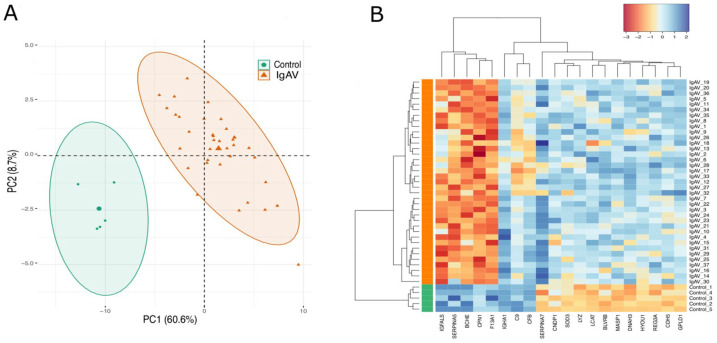
PCA and heatmap of proteome data. (**A**): PCA of the proteome data in a 2D graph of PC1 and PC2. The biplot shows proteome data (LFQ intensities) as labeled dots and triangles and group effect (loadings) as vectors. (**B**): Heatmap presentation of a hierarchical cluster of the 20 proteins that show significantly different (*p* ≤ 0.05 and FC < −1.5 or FC >1.5) LFQ intensities in both groups (IgAV (I) and control (C) strain) at differential find proteins. The red color represents low, and the blue color represents high expressed protein levels.

**Figure 2 diagnostics-13-01729-f002:**
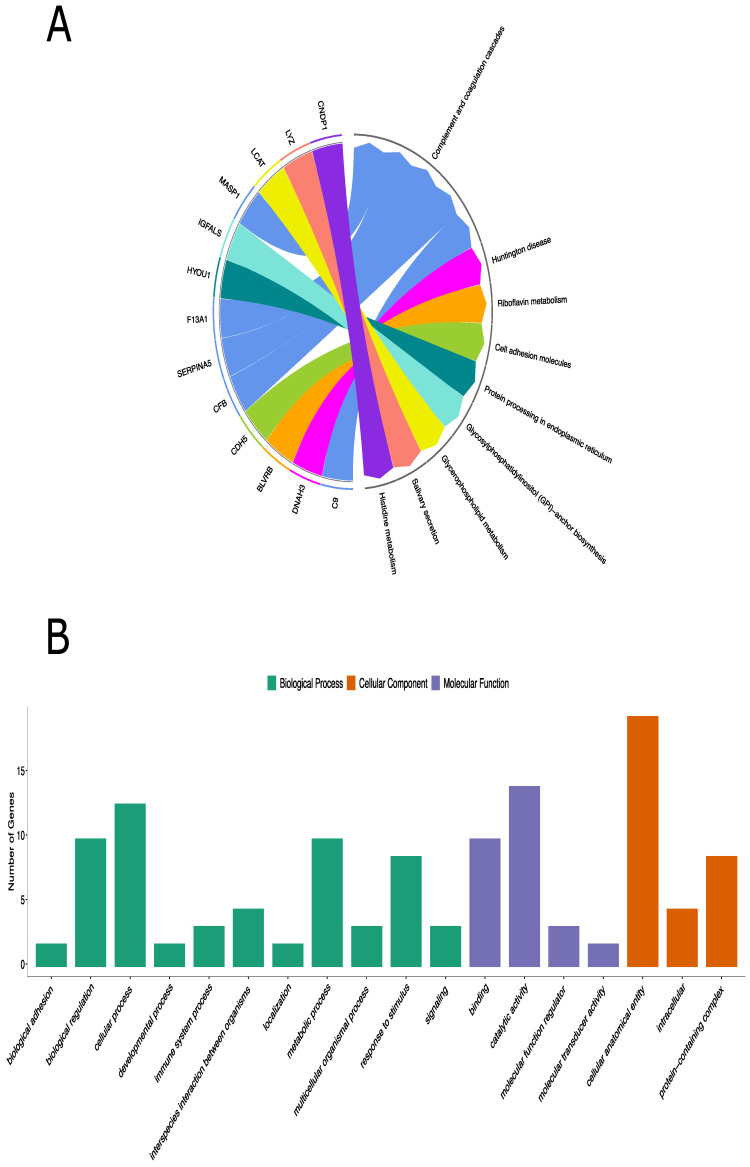
KEGG pathway analysis and GO annotation analyses of the differentially expressed proteins in children with IgAV. (**A**): KEGG database defined 8 different pathways for 13 out of 20 differentially expressed proteins. (**B**): Functional classifications of identified proteins using PANTHER gene classification system. The abscissa represents the GO annotation classifications, which are divided into three major categories: biological process (BP), molecular function (MF), and cellular component (CC). The ordinate represents the number of different proteins under each functional classification.

**Figure 3 diagnostics-13-01729-f003:**
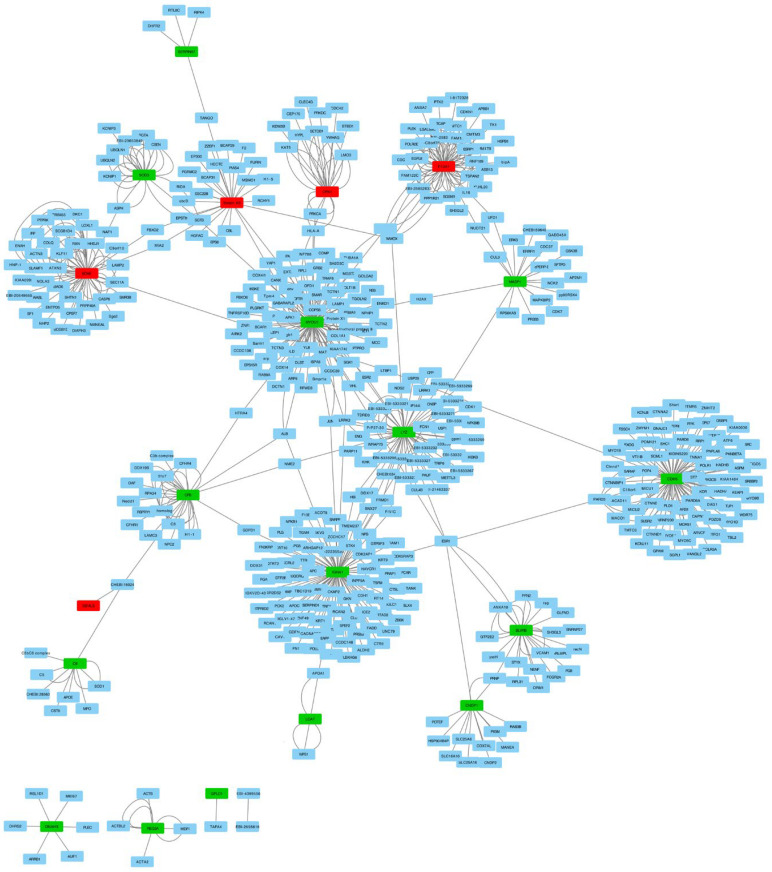
Network analyses of the 20 differentially expressed proteins and their interactions in children with IgAV using Cytoscape. Red nodes present downregulated genes and green nodes represent upregulated genes. Blue nodes are the 493 protein interactions found in the IntAct molecular interaction database.

**Figure 4 diagnostics-13-01729-f004:**
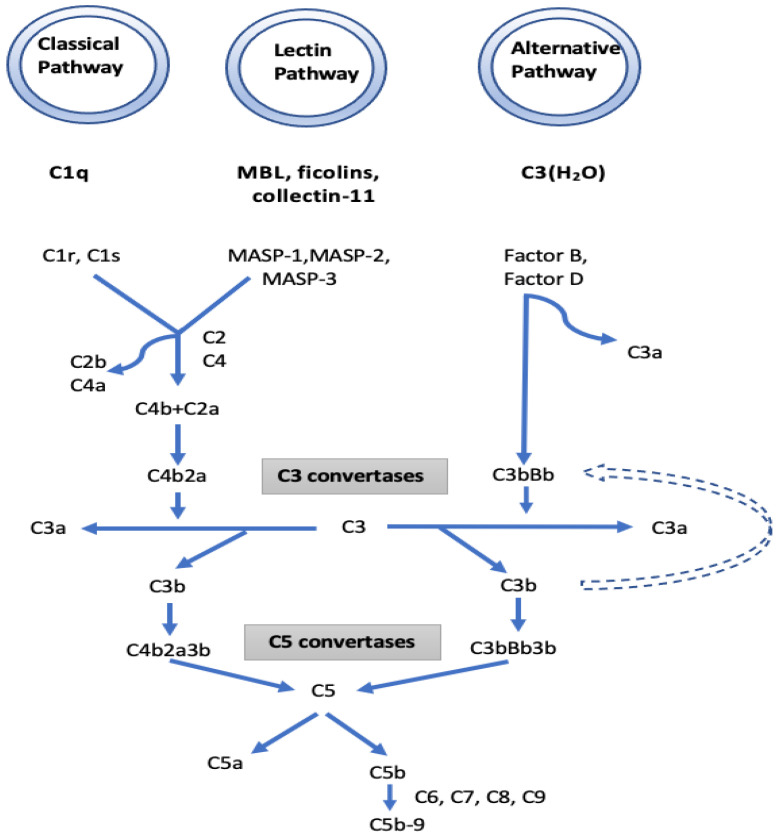
The initiators, components, and regulators of the different complement pathways. Mannose-binding lectin-associated serine protease 1 (MASP1), which is a component of the lectin pathway, and complement factor B, which is a component of the alternative pathway, have been found to increase in patients with IgAV compared to the control group.

**Table 1 diagnostics-13-01729-t001:** Main KEGG pathways and functions of the differentially expressed proteins in children with IgAV.

Gene Name	Protein Name	IgAV vs. Control (Fold Change)	KEGG Pathway	Function
MASP1	Mannose-binding lectin-associated serine protease 1	13.46	Complement and coagulation cascades	Lectin pathway of complement activation/coagulation pathway
CFB	Complement factor B	1.72	Complement and coagulation cascades	Alternate pathway of complement activation
SERPINA5	Plasma serine protease inhibitor	−3.7	Complement and coagulation cascades	Inhibit several serine proteases; hemostasis/thrombosis/complement activation pathways
C9	Complement C9	1.99	Complement and coagulation cascades	Constituent of membrane attack complex (MAC)
SERPINA7	Thyroxine-binding globulin	19.46		
FXIIIA1 (F13A1)	Coagulation factor XIII a-chain	−5	Complement and coagulation cascades	Fibrin stabilizing
REG3A	Regenerating family member 3 alpha	14.29		Innate Immune system
GPLD1	Glycosylphosphatidylinositol-specific phospholipase D1	14.72		
LCAT	Lecithin–cholesterol acyltransferase	12.95	Glycerophospholipid metabolism	
SOD3	Superoxide dismutase 3	13.69		Antioxidant
HYOU1	Hypoxia upregulated 1	10.28	Protein processing in endoplasmic reticulum	Hypoxia chaperone
DNAH3	Dynein axonemal heavy chain 3,	13.65	Huntington disease	
CDH5	Cadherin 5	13.67	Cell adhesion molecules	Endothelial adherens junction assembly and maintenance
IGFALS	Insulin-like growth factor-binding protein acid labile subunit	−4.28	Glycosylphosphatidylinositol (GPI)-anchor biosynthesis	Cell adhesion
CNDP1	Carnosine dipeptidase 1	12.19	Histidine metabolism	Beta-alanine metabolism
BCHE	Butyrylcholinesterase	−5.08		
CPN1	Carboxypeptidase N subunit 1	−2.84		Anaphylatoxin inactivator; protects the body from potent vasoactive and inflammatory peptides
LYZ	Lysozyme C	13.72	Salivary secretion	
BLVRB	Flavin reductase	10.79	Riboflavin metabolism	Heme metabolism
IGHA1	Immunoglobulin heavy constant alpha 1	1.98		Immune system, immunoglobulin receptor binding

## Data Availability

The data presented in this study are available on request from the corresponding author.

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
