# Peer review of "Plasma Proteomic Analysis Reveals the Potential Role of Lectin and Alternative Complement Pathways in IgA Vasculitis Pathogenesis"

_diagnostics, 2023, doi:10.3390/diagnostics13101729_

Round 1

Reviewer 1 Report

IgAV is a small-vessel form of the autoimmune vasculitis caused by IgA1-mediated inflammation. The underlying mechanism of IgAV and its most common and severe complication, IgA vasculitis with nephritis, remains unclear. In this study, the authors use an untargeted label-free approach to investigate the alterations in plasma proteomic profiles of IgAV pediatric patients without renal involvement. The study had two main objectives. Firstly, to identify and quantify proteins to assist diagnosis and treatment and secondly provide new insight on the possible underlying molecular mechanism of IgAV progression. 42 children were enrolled, including 37 IgAV patients and 5 healthy controls. A total of 25 proteins were differentially expressed in non-depleted plasma samples of IgAV patients relative to controls. These proteins are involved in the modulation of multiple physiological process and pathways, including -but not limited to- the lectin and alternative pathways of the complement, blood coagulation, antigen recognition and binding, and peptidase activity. Pathway enrichment analysis conducted on those differentially expressed proteins revealed several pathways associated with IgAV. Based upon the result obtained the authors propose that i) the lectin and alternative complement pathways are involved in the pathogenesis of IgAV and ii) there is a prominent crossroad between complement and coagulation systems.

One (potential) strength of this paper is that MS data were acquired with a novel data-independent acquisition (DIA) method. In principle, DIA may address some of the limitations of the traditional data-dependent acquisition (DDA) mass spectrometry in plasma.

Unfortunately, the data presented in support of the above suggestions are insufficient and/or do not support many of the conclusions and claims made.

One issue of particular concern for this reviewer are the results discussed under ‘Immunoglobulin related proteins’ that were not interpreted appropriately.

My comments are listed below in the order they appear in the text rather than order of importance.

Specific comments

Abstract, line 24 & Results, line 129 and 130

The authors claim that 418 proteins were identified (line 24: “Among the 418 proteins identified…”) but also that they obtained the expression profiles for 418 proteins (lines 129 and 130: “data resulting in expression profiles for 418 proteins in the plasma from patients with IgAV…”).

Please clarify: how many proteins were identified and how many proteins were quantified?

Materials and Methods, line 78

…five age- and gender-matched healthy controls

IgAV patients were age-matched with healthy controls (8.77±3.54 vs 8.91±4.07 years, respectively). However, they were not gender-matched, as indicated in the text. The ratio of male to female was 0.54:1 vs 1.5:1, for IgAV and HC, respectively.

Results, line 142

Add 3.3. in front of “KEGG and Reactome pathway analysis of the differentially expressed proteins” to indicate that this is a subtitle.

Figure 1B, Heatmap, line 143

The heatmap contains 26 columns but there are only 25 proteins that show significantly different LFQ intensities. Where is 26th protein coming from?

One column in the heatmap (fifth column starting from the left) is entitled ‘NA’. Please replace ‘NA’ by the appropriate protein name.

Please insert a color key above heatmap that includes Z score values.

Figure 1, legend, line 147

FC < -1.5 or FC > 1.5, not LFC < -1.5 or LFC > 1.5, as indicated in the text

Figure 1, legend, line 148

The legend to Figure 1B is misleading. It is written that “The green color represents low and the pink color represents high expressed protein levels”. This is incorrect. In the heatmap image, red color represents downregulated proteins, and blue color, represents upregulated proteins. The leftmost column is colored according to group (green = control, orange = IgAV patients).

Results, line 150

KEGG pathway analysis includes 12 of 25 differentially expressed proteins, not 13, as indicated in the text.

Results, line 152

…the Reactome pathway analysis identified 18 out of 25 differentially expressed proteins, not 19, as indicated in the text.

Figure 2, legend, line 162

The ordinate represents the number of differentially expressed proteins, not the number of different proteins, as written in the text.

Table 1, line 164

Most likely, the five Fab peptides identified by the authors have incorrectly been identified and quantified (see additional comments below)

Results, line 166

IgHA1 was expressed which is linked to the glycosylation of IgA and in the response to elevated platelet calcium

This statement is not supported by any experimental data and appears to be far-fetched.

Results, line 171

Add 3.4. in front of “Panther classification analysis and GO functional analysis of the differentially expressed proteins” to indicate that this is a subtitle.

Discussion, line 227

Mesangial deposition of C4d is not a good marker of activation of the lectin pathway, as C4d is also a degradation product of the classic pathway.

Discussion, line 277

Results discussed under ‘Immunoglobulin related proteins’ were not interpreted appropriately.

Database searching of mass spectrometry data is not a suitable approach for determining qualitative (presence/absence) and quantitative (intensity) differences in antigen binding fragment (Fab) of immunoglobulins. The Uniprot database (UP000005640) used in this study contains 15 sequences of IGHV1 germline gene origin, 35 sequences of IGHV3 germline gene origin, and 11 sequences of IGHV4 germline gene origin.  Each of the above sequence includes two hypervariable complementary-determining regions (CDR1 and 2) and three structurally conserved framework region (FR1, 2, and 3). The sequences having the same germline gene origin exhibit very high sequence homology/identity in their framework region and are therefore difficult to distinguish from each other by Mass spectrometry. Importantly, only a small fraction of the possible sequences is represented in the Uniprot database and hence most will not be identified by interrogating the database.

Most likely, the five Fab peptides identified by the authors have incorrectly been identified and quantified. Selection of ‘razor and unique peptides’ for the quantification using MaxQuant could be compounding the problem.

Supplementary Materials, line 351

Were MS data acquired with a library-based or library-free data-independent acquisition method?

Author Response

Reviewer 1

IgAV is a small-vessel form of the autoimmune vasculitis caused by IgA1-mediated inflammation. The underlying mechanism of IgAV and its most common and severe complication, IgA vasculitis with nephritis, remains unclear. In this study, the authors use an untargeted label-free approach to investigate the alterations in plasma proteomic profiles of IgAV pediatric patients without renal involvement. The study had two main objectives. Firstly, to identify and quantify proteins to assist diagnosis and treatment and secondly provide new insight on the possible underlying molecular mechanism of IgAV progression. 42 children were enrolled, including 37 IgAV patients and 5 healthy controls. A total of 25 proteins were differentially expressed in non-depleted plasma samples of IgAV patients relative to controls. These proteins are involved in the modulation of multiple physiological process and pathways, including -but not limited to- the lectin and alternative pathways of the complement, blood coagulation, antigen recognition and binding, and peptidase activity. Pathway enrichment analysis conducted on those differentially expressed proteins revealed several pathways associated with IgAV. Based upon the result obtained the authors propose that i) the lectin and alternative complement pathways are involved in the pathogenesis of IgAV and ii) there is a prominent crossroad between complement and coagulation systems.

One (potential) strength of this paper is that MS data were acquired with a novel data-independent acquisition (DIA) method. In principle, DIA may address some of the limitations of the traditional data-dependent acquisition (DDA) mass spectrometry in plasma.

Unfortunately, the data presented in support of the above suggestions are insufficient and/or do not support many of the conclusions and claims made.

One issue of particular concern for this reviewer are the results discussed under ‘Immunoglobulin related proteins’ that were not interpreted appropriately.

My comments are listed below in the order they appear in the text rather than order of importance.

Specific comments

  1. Abstract, line 24 & Results, line 129 and 130

The authors claim that 418 proteins were identified (line 24: “Among the 418 proteins identified…”) but also that they obtained the expression profiles for 418 proteins (lines 129 and 130: “data resulting in expression profiles for 418 proteins in the plasma from patients with IgAV…”).

Please clarify: how many proteins were identified and how many proteins were quantified?

Response: Thank you for pointing out the miswording of this sentence. Four hundred and eighteen proteins were identified and quantified using LFQ in Maxquant. We’ve changed the wording of this sentence now as: ‘Among the 418 proteins identified in the Nano LC-MS/MS analysis, 26 had significantly different expressions in IgAV patients. Among them, 20 were upregulated and 5 were downregulated.’

  1. Materials and Methods, line 78

…five age- and gender-matched healthy controls

IgAV patients were age-matched with healthy controls (8.77±3.54 vs 8.91±4.07 years, respectively). However, they were not gender-matched, as indicated in the text. The ratio of male to female was 0.54:1 vs 1.5:1, for IgAV and HC, respectively.

Response: Thank you for your comment, we removed the gender matched part of the sentence as below.

‘Thirty-seven active IgAV patients, and five age matched healthy controls were enrolled in the study.’

  1. Results, line 142

Add 3.3. in front of “KEGG and Reactome pathway analysis of the differentially expressed proteins” to indicate that this is a subtitle.

Response: We have added 3.3 in front of “KEGG and Reactome pathway analysis of the differentially expressed proteins”.

  1. Figure 1B, Heatmap, line 143
  2. The heatmap contains 26 columns but there are only 25 proteins that show significantly different LFQ intensities. Where is 26thprotein coming from?
  3. One column in the heatmap (fifth column starting from the left) is entitled ‘NA’. Please replace ‘NA’ by the appropriate protein name.
  • Please insert a color key above heatmap that includes Z score values.

Response: Thank you for your suggestion. Basically, of the 26 proteins one of them is unknown (P01825). So it was recorded as NA on the heatmap plot and kept outside of the bioinformatic analyses. Now, we’ve changed 25 to 26 throughout the text and changed the heatmap labeling (NA to P01825) to make it clear. New Figure 1B is created for this purpose including the color key as below:

  1. Figure 1, legend, line 147

FC < -1.5 or FC > 1.5, not LFC < -1.5 or LFC > 1.5, as indicated in the text

Response: Thank you, we have revised it according to your suggestion.

  1. Figure 1, legend, line 148

The legend to Figure 1B is misleading. It is written that “The green color represents low and the pink color represents high expressed protein levels”. This is incorrect. In the heatmap image, red color represents downregulated proteins, and blue color, represents upregulated proteins. The leftmost column is colored according to group (green = control, orange = IgAV patients).

Response: Thank you, we had used terrain coloring in the first version of the plot and forget to change it. Now it is corrected as red and blue.

  1. Results, line 150

KEGG pathway analysis includes 12 of 25 differentially expressed proteins, not 13, as indicated in the text.

Response: Thank you for you correction. Table 1 included 13 but now we have included it in the new Figure 2 as well.

  1. Results, line 152

…the Reactome pathway analysis identified 18 out of 25 differentially expressed proteins, not 19, as indicated in the text.

Response: Thank you for highlighting this typo, it is corrected as 18.

  1. Figure 2, legend, line 162

The ordinate represents the number of differentially expressed proteins, not the number of different proteins, as written in the text.

Response: Thank you,  we edited the sentence as below:

‘The ordinate represents the number of differentially expressed proteins under each functional classification.’

  1. Table 1, line 164

Most likely, the five Fab peptides identified by the authors have incorrectly been identified and quantified (see additional comments below)

  1. Results, line 166

IgHA1 was expressed which is linked to the glycosylation of IgA and in the response to elevated platelet calcium

This statement is not supported by any experimental data and appears to be far-fetched.

Response: This statement was removed it from the results section. In the discussion section heavy, lambda variable regions and constant regions of the immunoglobulins linked to aberrant glycosylation are discussed with references.  

  1. Results, line 171

Add 3.4. in front of “Panther classification analysis and GO functional analysis of the differentially expressed proteins” to indicate that this is a subtitle.

Response: Thank you, we added it.

  1. Discussion, line 227

Mesangial deposition of C4d is not a good marker of activation of the lectin pathway, as C4d is also a degradation product of the classic pathway.

Response: It is true that C4d alone is not a marker of the activation of the lectin pathway. Mesangial deposition of C4d is not rare in IgAN and was initially considered to be a consequence of classical pathway activation. However, recent literature acknowledges that these C4 deposits are probably due to activation through the lectin pathway since the co-deposition of C4d with MBL–associated serine proteases (MASPs; MASP-1 and MASP-2), l-ficolin (references 20 and 26). Initially C4 (and C4d) and C4-binding protein were thought to be markers of classical pathway activation, in nowadays they are believed to be more likely products of activation of the lectin pathway.

  1. Discussion, line 277

Results discussed under ‘Immunoglobulin related proteins’ were not interpreted appropriately.

Database searching of mass spectrometry data is not a suitable approach for determining qualitative (presence/absence) and quantitative (intensity) differences in antigen binding fragment (Fab) of immunoglobulins. The Uniprot database (UP000005640) used in this study contains 15 sequences of IGHV1 germline gene origin, 35 sequences of IGHV3 germline gene origin, and 11 sequences of IGHV4 germline gene origin.  Each of the above sequence includes two hypervariable complementary-determining regions (CDR1 and 2) and three structurally conserved framework region (FR1, 2, and 3). The sequences having the same germline gene origin exhibit very high sequence homology/identity in their framework region and are therefore difficult to distinguish from each other by Mass spectrometry. Importantly, only a small fraction of the possible sequences is represented in the Uniprot database and hence most will not be identified by interrogating the database.

Most likely, the five Fab peptides identified by the authors have incorrectly been identified and quantified. Selection of ‘razor and unique peptides’ for the quantification using MaxQuant could be compounding the problem.

Response: We would like to thank you for this discussion. To clarify, in the Protein identification area, one can decide to only use unique peptides in the relative protein quantification analysis. But the default value is use razor and unique peptides. So, we’ve selected this for the quantification of the protein group with a larger number of associated peptides (unique + razor peptides). We’ve downloaded the up-to-date protein fasta from Uniprot because UniProt database is the world's leading high-quality, comprehensive and freely accessible resource of protein sequence and functional information. We’ve selected three recent studies from 2022 to show the Uniprot Homo sapiens (Proteome ID: UP000005640) reference proteome fasta is widely used in proteomics studies.

Mass spectroscopy-based proteomics and metabolomics analysis of triple-positive breast cancer cells treated with tamoxifen and/or trastuzumab.

Sharaf BM, Giddey AD, Al-Hroub HM, Menon V, Okendo J, El-Awady R, Mousa M, Almehdi A, Semreen MH, Soares NC.Cancer Chemother Pharmacol. 2022 Dec;90(6):467-488. doi: 10.1007/s00280-022-04478-4.

Multi-Omics Analysis Revealed a Significant Alteration of Critical Metabolic Pathways Due to Sorafenib-Resistance in Hep3B Cell Lines.

Abushawish KYI, Soliman SSM, Giddey AD, Al-Hroub HM, Mousa M, Alzoubi KH, El-Huneidi W, Abu-Gharbieh E, Omar HA, Elgendy SM, Bustanji Y, Soares NC, Semreen MH.Int J Mol Sci. 2022 Oct 9;23(19):11975. doi: 10.3390/ijms231911975.

SARS-CoV-2-positive patients display considerable differences in proteome diversity in urine, nasopharyngeal, gargle solution and bronchoalveolar lavage fluid samples.

Okendo J, Musanabaganwa C, Mwangi P, Nyaga M, Onywera H.PLoS One. 2022 Aug 8;17(8):e0271870. doi: 10.1371/journal.pone.0271870. eCollection 2022.PMID: 35939435

  1. Supplementary Materials, line 351

Were MS data acquired with a library-based or library-free data-independent acquisition method?

Response: Data independent acquisition (DIA) method was used to acquire MS data.

Reviewer 2 Report

In this study, authors performed a nano liquid chromatography-tandem mass spectrometry in IgAV patient in order to better appreciate IgAV pathophysiology.

The subject is new and interesting. Graph and tables are very well presented and this study brings new interesting data to scientific community.

I have only one question :

Do you have biological data for these patients and more particulary complement (for example sC5B9) and coagulation proteins anaysis ?

Author Response

Response: Thank you for your comment, unfortunately we do not have any biological data of those patients for coagulation protein analysis. Complement 3 levels were available in 19 patients and were within normal range.

Round 2

Reviewer 1 Report

Points 1 to 13 have been addressed appropriately in the revised version of the manuscript submitted by the authors. However, point 14 was not (results discussed under ‘Immunoglobulin related proteins’). Most likely, the five-antigen binding fragment (Fab) peptides identified by the authors have incorrectly been identified and quantified, as already explained in my first review. Therefore, this reviewer believes that the manuscript still contains a significant flaw and should not be accepted in its present form.

To render the manuscript acceptable for publication the authors need to remove all passage in the text that mention the variable domain of immunoglobulins:

They should remove the first five entries in Table 1, (line 166)

IGHV3-74 Immunoglobulin Heavy Variable 3-74

IGLV4-69 Immunoglobulin Lambda Variable 4-69

IGHV1-69-2 Immunoglobulin Heavy Variable 1-69-2

IGHV4-31 Immunoglobulin Heavy Variable 4-31

IGHV1-24 Immunoglobulin Heavy Variable 1-24

They should remove all the results discussed under ‘Immunoglobulin related proteins’, e.g. remove line 279 to 295.

Author Response

Dear reviewer,  We've understood your concern, and since it is valid, we decided to remove those which might be false positive results. Now we have removed the IGHVX and unknown Ig-like proteins from the study. The manuscript is updated accordingly including figures, tables, and supplementary material. 

Round 3

Reviewer 1 Report

Point 14 has been addressed appropriately in the revised version of the manuscript.